# Epidemiology of Zoonotic *Coxiella burnetii* in The Republic of Guinea

**DOI:** 10.3390/microorganisms11061433

**Published:** 2023-05-29

**Authors:** Olesia V. Ohlopkova, Sergey A. Yakovlev, Kabwe Emmanuel, Alexey A. Kabanov, Dmitry A. Odnoshevsky, Mikhail Yu. Kartashov, Alexey D. Moshkin, Igor V. Tuchkov, Nikita Yu. Nosov, Andrey A. Kritsky, Milana A. Agalakova, Yuriy N. Davidyuk, Svetlana F. Khaiboullina, Sergey P. Morzunov, Magasuba N’Fally, Sanaba Bumbali, Mamadou Fode Camara, Mamadou Yero Boiro, Alexander P. Agafonov, Elena V. Gavrilova, Rinat A. Maksyutov

**Affiliations:** 1State Research Center of Virology and Biotechnology «Vector» of Rospotrebnadzor, Koltsovo 630559, Russia; 2Russian Research Anti-Plague Institute «Microbe» of Rospotrebnadzor, Saratov 410005, Russia; 3OpenLab “Gene and Cell Technologies”, Institute of Fundamental Medicine and Biology, Kazan (Volga Region) Federal University, Kazan 420008, Russia; 4State Research Center of Dermatovenerology and Cosmetology of Russian Ministry of Health, Moscow 107076, Russia; 5Limited Liability Company, «Biotech Campus», Moscow 117437, Russia; 6Faculty of Preventive Medicine, Ural State Medical University, Yekaterinburg 620014, Russia; 7Limited Liability Company, «Quality Med», Yekaterinburg 105318, Russia; 8Department of Pathology, University of Nevada, Reno, NV 89557, USA; 9Faculty of Medicine, Pharmacy and Dentistry, University Gamal Abdel Nasser, Conakry 001, Guinea; 10Research Institute of Applied Biology of Guinea, Kindia 100, Guinea; 11Higher Institute of Science and Veterinary Medicine, Dalaba 280, Guinea

**Keywords:** infectious disease, Q fever, ticks, zoonosis

## Abstract

Background: Q fever is a zoonotic infectious disease characterized by fever, malaise, chills, significant weakness, and muscle pain. In some cases, the disease can become chronic and affect the inner membranes of the heart, such as the valves, leading to endocarditis and a high risk of death. *Coxiella burnetii* (*C. burnetii*) is the primary causative agent of Q fever in humans. This study aims to monitor the presence of *C. burnetii* in ticks collected from small mammals and cattle in the Republic of Guinea (RG). Methods: Rodents were trapped in the Kindia region of RG during 2019–2020, and ticks were collected from cattle in six regions of RG. Total DNA was extracted using a commercial kit (RIBO-prep, InterLabService, Russia) following the manufacturer’s instructions. Real-time PCR amplification was conducted using the kit (AmpliSens Coxiella burnetii-FL, InterLabService, Russia) to detect *C. burnetii* DNA. Results and Conclusions: Bacterial DNA was detected in 11 out of 750 (1.4%) small mammals and 695 out of 9620 (7.2%) tick samples. The high number of infected ticks (7.2%) suggests that they are the main transmitters of *C. burnetii* in RG. The DNA was detected in the liver and spleen of a Guinea multimammate mouse, *Mastomys erythroleucus*. These findings demonstrate that *C. burnetii* is zoonotic in RG, and measures should be taken to monitor the bacteria’s dynamics and tick prevalence in the rodent population.

## 1. Introduction

*Coxiella burnetii* is an obligate intracellular parasite and a member of the *Coxiellaceae* family [1]. Although this bacterium shares morphological similarities with Rickettsia, phylogenetic analysis has demonstrated that it belongs to a separate gamma subdivision of Proteobacteria [2,3]. *Coxiella burnetii* is a gram-negative, rod-shaped bacterium that replicates within eukaryotic cell vacuoles [4]. It exhibits high resistance to physical and chemical environmental factors [5], which contributes to its ability to cause human infection. Studies have shown that even after exposure to 1% phenol solution or 0.5% chloramine solution for more than three days, *C. burnetii* can remain viable [6]. Additionally, it can survive for up to 18 months in the dry feces of infected *Dermacentor andersoni* ticks [7]. However, boiling can kill the bacterium within 10 min in milk, whereas it can persist for up to a month in raw meat [8,9]. Furthermore, *C. burnetii* displays resistance to ultraviolet radiation and can persist at low temperatures. These findings suggest that *C. burnetii* can circulate in the environment for an extended period while maintaining a high level of virulence.

Acute Q fever typically presents as a mild or localized disease with a low risk of death. However, if left untreated, chronic Q fever can lead to mortality, primarily due to endocarditis or vascular infection. Patients treated for endocarditis caused by chronic Q fever have a 10-year mortality rate of 19%. Q fever shares clinical symptoms similar to those of various bacterial and viral infectious diseases [10]. The incubation period can last up to 60 days. The most common manifestation of the infection is an abrupt onset of fever, with body temperature rising to 39–40 °C, accompanied by chills, headache, dry cough, weakness, decreased appetite, and sleep disturbances [11]. Patients may also experience muscle and joint pain, dizziness, nausea, vomiting (less frequently), and nosebleeds. As the disease progresses, signs of organ involvement may emerge, and pneumonia may occur in 8–32% of cases [12]. Post-infectious asthenia is the most common complication of acute Q fever. Chronic coxiellosis affects 5–10% of patients. Pregnant women infected with *C. burnetii*, the causative agent of Q fever, may experience unfavorable pregnancy outcomes. The mortality rate for coxiellosis is less than 2.8% [13].

Humans can contract Q fever by inhaling contaminated aerosols or consuming contaminated animal products. Contaminated aerosols, which can contain *C. burnetii*, are generated during the processing of parturient fluids, placenta, or wool from infected animals [14]. While *C. burnetii* has been found in ticks, arthropod-borne transmission to humans is believed to play only a minor role in outbreaks [15]. Q fever is a global concern, often associated with cattle and sheep farming [16]. One of the largest documented outbreaks occurred in the Netherlands between 2007 and 2010, where over 4000 individuals became ill and 14 died [17,18].

Despite efforts by the World Health Organization (WHO) to reduce its spread and control morbidity and mortality, *C. burnetii* remains a public health problem in many countries [19]. Factors such as globalization, migration, and tourism can contribute to the spread of the disease [20]. In the Republic of Guinea (RG), low life expectancy is linked to various viral and bacterial diseases, including Lassa fever, West Nile virus, Crimean-Congo hemorrhagic fever, yellow fever, Chikungunya, Rift Valley fever, and Q fever [21]. According to the WHO report, the current life expectancy in RG is between 45 and 50 years [22]. Therefore, greater attention should be given to the prevention of these related diseases. One potential preventive measure could involve regular screening of rodents and ticks, as well as increasing anti-epidemic activities in high-risk areas for Q fever.

In nature, *C. burnetii* is found in more than 60 species of warm-blooded animals and approximately 50 species of birds, which serve as primary reservoirs [23]. Ticks that feed on infected animals are the primary carriers of *C. burnetii* [24]. Farm animals such as cattle, goats, and sheep typically transmit the bacteria to humans. *C. burnetii* can cause transient bacteremia in infected animals, during which feeding ticks can become infected [14]. There are two recognized types of natural Q fever infections: primary (natural) and secondary (agricultural or anthropogenic) [7]. In natural foci, pathogens are transmitted by ticks, primarily Ixodes ticks and, to a lesser extent, *Haemaphysalis* ticks. Rickettsia can persist in these ticks and is transmitted transovarially and transstadially (from egg, larva, nymph, to adult tick), making them not only carriers but also reservoirs of the Q fever pathogen [9]. The evidence demonstrating direct transmission of *C. burnetii* from ticks to humans remains limited. However, there have been a few reported cases describing the transmission of *C. burnetii* to humans through tick bites [25,26,27,28,29,30].

Morbidity primarily occurs sporadically and affects professional risk groups such as cattle breeders and rural inhabitants, mainly during the spring, summer, and autumn seasons. Epidemic outbreaks are also possible. Infection from an infected person is rare and typically occurs through exposure to contaminated sputum and milk from nursing women [5]. In agricultural areas, the sources of the infection agent are cattle, small ruminants, horses, pigs, dogs, poultry, and rodents. In natural environments, the sources are wild ungulates, small mammals (primarily rodents), and birds. Mammals such as cattle and pigs are of epidemiological significance as they excrete rickettsiae through excreta, urine, milk, and amniotic fluid [10].

People can become infected in agricultural areas through various routes, including exposure to contaminated air and dust when handling wool, fur, bristles, and animal skin; consumption of contaminated milk and milk products; and contact with infected animals during care, slaughter, and butchering. Secondary sources pose a higher risk for humans in settled areas compared to primary sources [31].

The objective of this study was to investigate the prevalence of *C. burnetii* in rodents and ticks in the Republic of Guinea (RG), where the circulation of this pathogen is poorly understood.

## 2. Materials and Methods

Small mammal and rodent samples: Between 2019 and 2020, 250 small mammals were captured on farms in the Kindia region of RG and separated by species. The geographical locations of rodent trapping sites are shown in Figure 1. Blood-sucking arthropods (ticks) were collected from 7600 farm animals (cattle) in the Kindia, Mamou, Faranah, Labe, Kankan, and N’Zerekore regions of RG from 2016 to 2020. A total of 9620 Ixodidae ticks of eight species were identified.

Animal rights compliance: The study was conducted in accordance with the requirements of Sanitary Rules 1.3.3118-13, titled “Safety of work with microorganisms of risk (hazard) groups 1 and 2” [32]. It adhered to the legislation of the Russian Federation as well as reference documents issued by the Federal State Budgetary Institution of Science and Research Center “Vector” of Rospotrebnadzor, Russia, and followed the International Ethical Standards. Euthanasia of animals was performed using cervical dislocation. Brain, lung, kidney, liver, and spleen samples were collected from each small mammal. Suspensions of the collected tissues and ticks were prepared by homogenization in saline solution and used immediately for DNA extraction.

DNA extraction: The collected samples were subjected to lysis using a solution (RIBO-prep, InterLabService, Moscow, Russia), which resulted in the breakdown of cellular membranes and other biopolymer complexes, releasing nucleic acids and cellular components. After the addition of a precipitation solution and centrifugation, the dissolved DNA precipitated, while other components of the lysed clinical material remained in solution and were removed through subsequent washing. In the final step of the extraction, the precipitate was dissolved in an elution buffer, resulting in purified DNA in solution. This procedure yielded a purified DNA preparation that was free from amplification reaction inhibitors, ensuring high analytical sensitivity for PCR analysis.

Real-time polymerase chain reaction (RT-PCR) was performed using the AmpliSens *Coxiella burnetii*-FL reagent kit (InterLabService, Moscow, Russia) following the manufacturer’s specifications. This test targets the groEL gene of the microbe and is specifically designed for the detection of C. burnetii [31,33,34]. The test has demonstrated no false-positive results when using a closely related bacterial DNA template [35]. According to the manufacturer’s instructions, total DNA was extracted from the suspensions using the RIBO-prep reagent kit (InterLabService, Moscow, Russia). All testing was conducted at the Russian-Guinean Scientific and Clinical Diagnostic Center for Epidemiology and Microbiology, located in Kindia, the Republic of Guinea (RG).

Statistical analysis: The prevalence index (PI) was calculated using the equation [36]:PI = (ni/N) × 100.(1)
where PI is the prevalence index; ni is the number of individuals; and N is the total number of individuals in the population. The abundance index (AI) was calculated using the equation [37]: AI = m/N.(2)
where m is the number of ticks and N is the number of individuals. Statistical analysis was performed in the R environment [38]. Statistically significant differences between comparison groups were accepted as *p* < 0.05, as assessed by the Kruskal-Wallis test with Benjamini-Hochberg (BH) adjustment for multiple comparisons. Correlations were analyzed using the R psych package, and *p*-values were adjusted with the Benjamini-Hochberg method based on Spearman’s rank correlation coefficient.

## 3. Results

### 3.1. Coxiella Burnetii DNA Prevalence in Small Mammal Tissue

We used 750 small mammal tissue samples in our study, identified as black rats (*Rattus rattus)* (73 mammals), Guinea multimammate mice (*Mastomys erythroleucus*) (85 mammals), Natal multimammate mice (*Mastomys natalensis*) (27 mammals), Guinean gerbil (*Gerbilliscus guinea*) (4 mammals), Dalton’s mouse (*Praomys daltoni*) (6 mammals), Temminck’s mouse (*Mus musculoides*) (16 mammals), house mouse (*Mus musculus*) (28 mammals), Sudanian grass rat (*Arvicanthis androgen*) (4 mammals), and the African giant shrew (*Crocidura olivieri*) (11 mammals). Five organs (brain, spleen, liver, kidney, and lungs) were obtained from each small mammal. *Coxiella burnetii* DNA was detected in the organs of 11 captured mammals (Table 1).

*Coxiella burnetii* DNA was mainly detected in the liver, spleen, lungs, and kidney among the tissues studied. The bacterial DNA was frequently found in the *M. erythroleucus* rodent species, with a PI of 8.2%. In addition, *C. burnetii* DNA was detected in a single organ sample collected from *G. guineae*, *M. musculus,* and *R. rattus*. The prevalence index for the detection of bacterial DNA did not exceed 9.1% in these species. 

Bacterial nucleic acid was also isolated from ticks of the *Laelapidae* family that were collected from small mammals. Interestingly, a significant number of *C. burnetii* DNA-positive ticks were found on *M. erythroleucus*, which also exhibited the highest prevalence index for bacterial DNA detection in its organs. These findings suggest a potential role for ticks in the transmission of *C. burnetii* among these small mammals.

The small mammals were trapped in six different natural and satellite biotypes, including savannah, areas near water sources, agroecosystems, houses, stores/warehouses, and coniferous forests (Table 2). Small mammals that tested positive for *C. burnetii* DNA were identified in three biotypes: savannah (83.5%), stores/warehouses (9.7%), and agroecosystems (6.8%). It is noteworthy that the majority of small mammals with detectable bacterial DNA were captured in wilderness areas.

### 3.2. Coxiella burnetii DNA Prevalence in Ticks Collected from Cattle

We conducted an examination of 303 cattle to assess the presence of blood-sucking arthropods during both the dry and rainy seasons. Overall, we collected a total of 9620 ticks from the cattle. The presence of *C. burnetii* DNA was observed in 563 ticks collected during the dry season and 132 ticks collected during the rainy season, as shown in Table 3.

### 3.3. Abundance of Ticks in Investigated Cattle

Ixodid ticks were the most dominant species found in 220 (72.6%) cattle. Out of the 695 positive ixodid ticks, the most frequently found arthropods were *Amblyomma variegatum* (544; 78.2%), *Rhipicephalus decoloratus* (62; 8.9%), *Rhipicephalus annulatus* (38; 5.4%), *Hyalomma truncatum Koch* (26; 3.7%), *Rhipicephalus sanguineus* (16; 2.3%), *Rhipicephalus* (*Boophilus*) *geigyi* (7; 1%), and *Haemophysalis leachi* (2; 0.28%) species (Table 4).

The AI, representing the number of ticks per animal, ranged from 2.1 in the dry season to 3.5 in the rainy season. These data indicate seasonal variations in tick abundance. It appears that there are more ticks on cattle during the rainy season as compared to the dry season. 

Variations in tick species PI were found between the rainy and dry seasons. *H. leachi*, *Am. variegatum*, *Hy. truncatum*, *Rh. annulatus*, and *Rh. sanguineus* ticks were commonly found during the dry season. However, these species were less frequent during the rainy season. Instead, *Rh. decoloratus* and *Rh. geigyi* were most frequently found during the rainy season. It should be noted that *Am. variegatum* was the dominant tick species collected in both seasons.

Next, we analyzed the prevalence of *C. burnetii* in each tick species (Figure 1). Ticks of the following species were found to be positive for bacterium DNA in the dry season: *Am. variegatum*, *Hy. truncatum*, *Rh. geigyi*, *Baeph. Sp*, *Rh. annulatus*, *Rh. sanguineus*, *H. leachi*, and *Rh. decoloratus*. A smaller group of tick species, including *Am. variegatum*, *Hy. truncatum*, *Rh. geigyi*, and *Rh. decoloratus*, were positive in the rainy season.

## 4. Discussion

*Coxiella burnetii* is a gram-negative bacterium that resides within cells and is responsible for causing Q fever [39]. This infectious disease can be transmitted to humans through the consumption of contaminated products or the inhalation of aerosols containing the bacteria [40]. Ticks have been identified as susceptible to *C. burnetii*, and the bacteria are often found in the cells of their middle gut or stomach, which may explain their presence in tick feces [41]. Some studies have proposed ticks as potential biological vectors for transmitting the pathogen to mammals [42]. However, in recent years, a new member of the Coxiella genus called Coxiella-like endosymbiont (Coxiella-LE) has been discovered in ticks [43]. These microbes share morphological similarities with *C. burnetii* [44] and are commonly found in ticks [45], which complicates our understanding of the tick’s role in transmitting *C. burnetii*. Notably, the LE microbes lack virulence factors, suggesting a lower likelihood of being pathogenic [46]. Therefore, accurate identification of the *Coxiella* genus bacterium requires techniques like PCR and sequencing. In this study, we employed RT-PCR-based testing to determine the prevalence of *C. burnetii* in ticks collected from RG. This method exhibits high specificity in detecting *C. burnetii* DNA, ensuring accurate results.

Our findings align with previous studies that have reported the presence of *C. burnetii* in ticks in Africa [47] and contribute to our understanding of its circulation in RG. Previously, epidemiological data on this bacterium in RG had been limited to the detection of bacterium-specific antibodies in humans and animals [39,48]. Furthermore, we confirm earlier observations that *C. burnetii* can be found in ticks collected from small mammals [7,49]. The prevalence index (PI) we observed (7.2%) was higher than that reported in a systematic review of bacterial prevalence in hard ticks in Europe [50]. It appears that the prevalence of *C. burnetii* varies significantly across different regions. For instance, Ni et al. found *C. burnetii* in 55.66% of ticks [51], while Gonzalez et al. reported a high prevalence of 55.66% in the Meso-Mediterranean ecosystem [52]. A study conducted in Rome documented a *C. burnetii* prevalence of 22% in ticks [53]. In a meta-analysis by Yessinou et al., variations in *C. burnetii* prevalence were observed between Africa, Europe, and the Middle East [54]. In Africa, the prevalence of the bacteria in ticks ranged from 2.91% to 13.97%, which includes the value of 7.2% found in our study. Thus, it seems that the percentage of ticks carrying this human pathogen in RG is within the range commonly observed in other countries. The substantial variations in *C. burnetii* prevalence reported in different studies could be attributed to the gene targets used for PCR analysis, such as *IS1111*, *ompA*, and *icd* [55,56]. Similarly, Korner et al. reported variations in *C. burnetii* prevalence in ticks due to the gene target used for PCR analysis [57]. It has been demonstrated that the *icd* gene has high similarity between *C. burnetii* and Coxiella-LE, which could potentially lead to an overestimation of the tick’s role in pathogen transmission [58]. Additionally, factors such as climate, season, humidity, and the small number of ticks included in the study may contribute to the prevalence of *C. burnetii* in ticks. In a study by Troupin et al., a higher seroprevalence of *C. burnetii* was found in cattle in most of the studied prefectures [54]. Among cattle, seroprevalence was higher in females compared to males, and a higher frequency of antibody detection was observed in mature animals compared to young ones.

In our study, we employed a PCR method that specifically detects *C. burnetii* in ticks. This highly specific method was chosen to minimize the potential for false-positive results due to the presence of Coxiella-LE microbes. The discovery of Coxiella-LE microbes has prompted a re-evaluation of the role of ticks as vectors for *C. burnetii* [45]. Coxiella-LE is known to dominate in ticks and is believed to provide them with vitamin B [59]. Additionally, the genome of Coxiella-LE shares up to 97% identity with that of *C. burnetii* [60]. Such high identity in nucleic acid sequences can lead to misidentification.

To detect *C. burnetii*, several gene targets have been used, including *IS1111*, *icd*, *com1*, *sodB*, and *GroEL/htpB* [61,62,63]. However, some of these genes exhibit significant similarity between the bacterium and Coxiella-LE. Duron et al. demonstrated that approximately 30% of Coxiella-LE-carrying ticks tested positive for the IS1111 gene [64]. Furthermore, the *icd* sequence of Coxiella-LE shares 90% identity with the same gene in *C. burnetii* [58]. Studies have indicated that the most conserved gene, *groEL*, is the optimal target for *C. burnetii* detection using PCR methods [43,65]. In our study, we utilized the AmpliSens Coxiella burnetii-FL reagent kit, which targets the *groEL* gene. Therefore, the specificity of the PCR method employed in our study is sufficient to ensure accurate detection of *C. burnetii*. 

Our data reveals variations in the tick population on cattle, with Amblyomma being the most prevalent species, followed by Rhipicephalus and Hyalomma. These findings are consistent with previous observations made by Bayer and Maina [52]. We also observed that the tick abundance index (AI) was higher during the rainy season compared to the dry season. This supports the findings of Bayer and Maina, who demonstrated lower tick loads in dry seasons compared to rainy seasons in Nigeria [66]. Seasonal variations in tick abundance on cattle have been documented in other studies, such as the work by Babayani and Makati in Botswana [67]. It has been suggested that humidity supports tick survival and growth [68]. Therefore, the rainy season may pose a higher risk of tick-borne pathogen exposure for cattle.

Additionally, we observed changes in tick species between the two seasons. During the dry season, we collected more *H. leachi*, *Am. variegatum*, *Hy. truncatum*, *Rh. annulatus*, and *Rh. sanguineus* ticks. In contrast, these species were less frequent during the rainy season, while *Rh. decoloratus* and *Rh. geigyi* were more frequently found. Several factors have been identified to contribute to variations in tick numbers and species, including fur color, sex, and food preferences [69,70]. It has been demonstrated that males have fewer ticks than females during the dry season [67]. Additionally, cattle with darker fur tend to have more ticks compared to those with white fur. Another study found higher tick loads in calves during both dry and wet seasons, while seasonal changes were observed in adult calves, with fewer ticks during the dry season [71]. Interestingly, changes in food taste, such as sweet and sour, can affect tick species, as shown by Marufu et al. [72]. They reported that *Hyalomma* species were more prevalent in sour food. These changes in food taste could be influenced by seasonal variations in grass composition due to differences in water content.

We observed variations in the prevalence of *C. burnetii* in different tick species depending on the season. There were more tick species carrying the bacterium’s DNA during the dry season compared to the wet season. Limited data are available on *C. burnetii* prevalence during the dry and wet seasons in Africa. For instance, a study by Titcomb et al. in Kenya did not find any effect of rainfall on pathogen prevalence in ticks [73]. It is important to note that the method of tick collection can influence the interpretation of the results. Collecting ticks through dragging and flagging may not capture ticks that are actively feeding on animals [74]. Climate change can also impact the distribution of arthropod vectors. The tick population is dependent on the survival of larvae, which requires moisture [75]. Reductions in the duration and intensity of the rainy season [76,77] can affect tick survival. Additionally, the abundance of hosts is crucial for tick survival. Climate change, particularly longer dry seasons, can result in the migration of herbivores [25]. This could explain the higher species variations observed during the dry season. These findings suggest that a greater variety of tick species may contribute to the epidemiology of *C. burnetii* during the dry season.

We also discovered that ticks collected from small mammals tested positive for *C. burnetii* DNA. Interestingly, the highest prevalence of bacterial DNA was found in ticks collected from small mammals captured in the savannah. We also detected *C. burnetii* in ticks collected from small mammals in stores, warehouses, and agroecosystems, but the prevalence of bacterial DNA-positive mice was significantly lower. Notably, several organs of *M. erythroleucus*, a small mammal inhabiting the savannah, tested positive for bacterial DNA. These findings suggest that ticks may play a more significant role in the epidemiology of *C. burnetii* among wild small mammals.

## 5. Conclusions

Our findings significantly contribute to our understanding of the role of ticks in the epidemiology of *C. burnetii* in RG. This study represents the first analysis of *C. burnetii* epidemiology in ticks collected from both small mammals and cattle in the RG. We employed a highly specific PCR method to detect bacterial DNA, ensuring accurate identification. Our results revealed that multiple tick species can harbor the bacterium. Interestingly, we observed a higher prevalence of *C. burnetii* in ticks collected from small mammals inhabiting the savannah. This finding suggests that ticks may play a significant role in the epidemiology of *C. burnetii*. Furthermore, we noted a greater abundance of ticks collected from cattle during the wet season compared to the dry season. Additionally, we observed more variations in tick species during the dry season compared to the wet season. Similarly, a higher number of tick species tested positive for *C. burnetii* DNA in the dry season compared to the wet season.

## Figures and Tables

**Figure 1 microorganisms-11-01433-f001:**
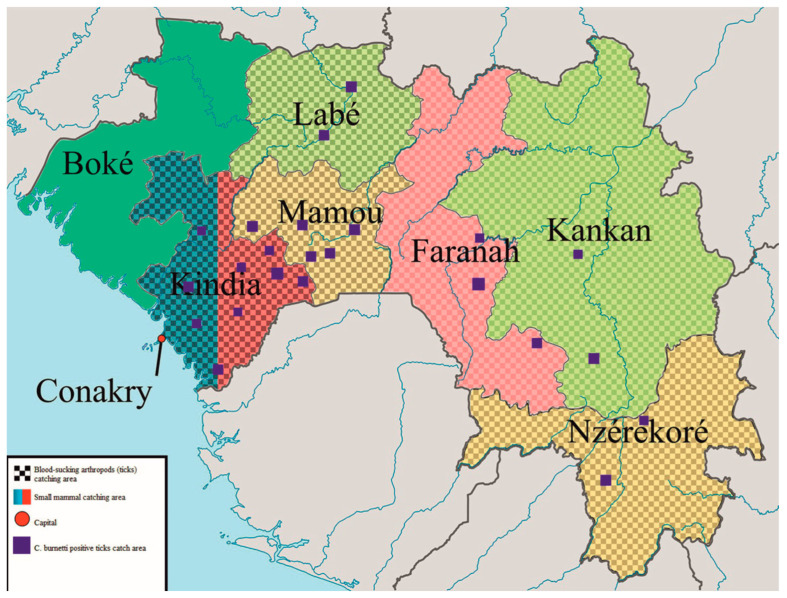
Shows the geographical locations of rodent trapping and blood-sucking arthropod (tick) collection sites.

**Table 1 microorganisms-11-01433-t001:** Small mammal species and prevalence index.

Small Mammals Species	Number of Small Mammals	PI (%)
*Mastomys natalensis*	1	1.3
*Mastomys erythroleucus*	7	8.2
*Gerbilliscus guineae*	1	25
*Rattus rattus*	1	1.3
*Mus musculus*	1	1.3
Total:	11	

**Table 2 microorganisms-11-01433-t002:** The distribution of traps in different biotypes set up in the RG areas.

Biotype	Setting Up the Trap (Times)	% Traps in Biotype	Number of Traps	% Traps in Biotypes/Total Traps	Small Mammals Captured (*n*)
Savannah	22	35.4	993	51.8	86
Neer water	1	1.6	7	0.4	0
Agrocenoses	5	8.1	199	10.4	16
Houses	25	40.3	600	31.3	143
Stores/warehouses	7	11.3	47	2,4	10
Coniferous forest	2	3.2	70	3.6	2
Total:	62		1916		250

**Table 3 microorganisms-11-01433-t003:** The abundance index by season shows the number of positive cattle for ticks and *C. burnetii* DNA-positive ticks.

Cattle (*n*)	Cattle Positive for Ticks	Number of Ticks	Yearly AI
Total during the period of study			
303	220	695	2.3
AI by season			
Dry season			
266	183	563	2.1
Rainy season			
37	37	132	3.5

**Table 4 microorganisms-11-01433-t004:** *Coxiella burnetii* DNA-positive Ixodid tick dominance indices by species composition and seasons.

Species	Total	PI	Rainy Season			Dry Deason		
			Total	PI	% Positive	Total	PI	% Positive
*Amblyomma ariegatum*	544	78.27	76	57.57	42.1	468	83.12	16.0
*Haemophysalis leachi*	2	0.28	0	0	0.0	2	0.35	50.0
*Hyalomma truncatum*	26	3.74	1	0.75	100.0	25	4.44	20.0
*Rhipicephalu annulatus*	38	5.46	0	0	0,0	38	6.74	26.3
*Rhipicephalus decoloratus*	62	8.92	48	36.36	52.1	14	2.48	28.6
*Rhipicephalus geigyi*	7	1.00	7	5.30	57.1	0	0	0.0
*Rhipicephalus sanguineus*	16	2.30	0	0	0.0	16	2.84	25.0
Total positive	695		132			563		

## Data Availability

All data are presented in the text.

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
