# Peer review of "Epidemiology of Zoonotic Coxiella burnetii in The Republic of Guinea"

_microorganisms, 2023, doi:10.3390/microorganisms11061433_

Round 1
Reviewer 1 Report
Dear authors,
Here are some suggestions:
- Line 25: the abbreviation of the Genus (Coxiella) is a stablished taxonomic rule and do not need to feature in parenthesis. Remove "(C. burnetii)" in the line.
- Line 34: the abbreviation of the Genus (Mastomys) is a stablished taxonomic rule and do not need to feature in parenthesis. Remove "(M.erythroleucus)" in the line.
- Line 37: All the keyword is in the title and it is not recommended. Suggestion: Infectious disease; Q-fever; ticks; zoonosis
- Line 40: the abbreviation of the Genus (Coxiella) is a stablished taxonomic rule and do not need to feature in parenthesis. Remove "(C. burnetii)" in the line.
- Line 49: Replace "[8].In contrast" with "[8]. In contrast".
- Line 67-78: This paragraph is repetitive and contains background necessary for the context of the study.
- Line 96: What is Argus and Hamas ticks? Would be Argas and Hyalomma (or Haemaphysalis)?
- In general, the introduction has several repeated parts that need to be better structured.
- Line 114-121: Remove" We conducted the first examination of C. burnetii epidemiology in ticks collected from both small mammals and cattle. Our findings showed the presence of bacterial DNA in various organs of M. erythroleucus, a small mammal commonly found in the savannah. We also identified several species of ticks as carriers of the bacterium. The number of ticks collected from cattle was higher during the wet season compared to the dry season, and more tick species were identified during the dry season. Moreover, we found more species of ticks positive for C. burnetii DNA during the dry season than during the wet season.". This text is results, discusion or other and not a part of objectives of the study.
- Line 125: Remove the term "Supplemental".
- Line 127-128: Ixodidae should not be italicized.
- Line 130: Rename Scheme 1 as Figure 1. In the figure, Legends are not comprensible due to a poor resolution.
- Line 159: "pi" must be in uppercase..
- Line 170-179: All the abbreviation in this paragraph does not necessary. This is very well stablished by taxonomic rules.
- Line 179: Remove the term "Supplemental"
- Line 180: Genus name in the first word of any paragraph does not be abbreviated.
- Line 186: Laelapide should not be italicized.
- Tables may well be inserted in the text rather than as supplementary material. Properly insert and renumber the tables.
- Line 187: Remove the term "Supplemental"
- Line 192: Remove the term "Supplemental"
- Line 199: Remove the term "Supplemental"
- Line 200-204: All the abbreviation in this paragraph does not necessary. This is very well stablished by taxonomic rules.
- Line 206: Remove the term "Supplemental"
- Line 209: Genus name in the first word of any paragraph does not be abbreviated.
- Tables should not have horizontal lines in each row.
- In Tables, scientific names should not be abbreviated, unless there are footnotes identifying the names. It is assumed that figures and tables are elements that must be understood independently of the main text.
- Legends of figures and tables are poor described.
- Line 213: This figure is not representative. A figure in columns with expressed confidence intervals made in the R statistical package would be much better represented.
- Figures need to be renumbered.
- Line 219: Genus name in the first word of any paragraph does not be abbreviated.
- Line 236: Genus name in the first word of any paragraph does not be abbreviated.
Minimal refinement edits are required.
Author Response
Our findings significantly contribute to our understanding of the role of ticks in the epidemiology of C. burnetii in RG. This study represents the first analysis of C. burnetii epidemiology in ticks collected from both small mammals and cattle in RG. We employed a highly specific PCR method to detect bacterial DNA, ensuring accurate identification. Our results revealed that multiple tick species can harbor the bacterium. Interestingly, we observed a higher prevalence of C. burnetii in ticks collected from small mammals inhabiting the savannah. This finding suggests that ticks may play a significant role in the epidemiology of C. burnetii. Furthermore, we noted a greater abundance of ticks collected from cattle during the dry season compared to the wet season. Additionally, we observed more variations in tick species during the dry season compared to the wet season. Similarly, a higher number of tick species tested positive for C. burnetii DNA in the dry season compared to the wet season.
Dear authors,
Here are some suggestions:
- Line 25: the abbreviation of the Genus (Coxiella) is a established taxonomic rule and do not need to feature in parenthesis. Remove "(C. burnetii)" in the line.
Agree: the parenthesis was removed
- Line 34: the abbreviation of the Genus (Mastomys) is a stablished taxonomic rule and do not need to feature in parenthesis. Remove "(M.erythroleucus)" in the line.
Agree: the parenthesis was removed
- Line 37: All the keyword is in the title and it is not recommended. Suggestion: Infectious disease; Q-fever; ticks; zoonosis.
Agree: the keywords were replaced by the suggested keywords
- Line 40: the abbreviation of the Genus (Coxiella) is a stablished taxonomic rule and do not need to feature in parenthesis. Remove "(C. burnetii)" in the line.
Agree: the parenthesis was removed
- Line 49: Replace "[8].In contrast" with "[8]. In contrast".
Agree: the sentence was paraphrased (Studies have shown that even after exposure to 1% phenol solution or 0.5% chloramine solution for more than three days, C. burnetii can remain viable [6]. Additionally, it can survive for up to 18 months in the dry feces of infected Dermacentor andersoni ticks [7]. However, boiling can kill the bacterium within 10 minutes in milk, whereas it can persist for up to a month in raw meat [8][9]. Furthermore, C. burnetii displays resistance to ul-traviolet radiation and can persist at low temperatures). Lines 46-51
- Line 67-78: This paragraph is repetitive and contains background necessary for the context of the study.
Agree: the paragraph was edited (Humans can contract Q fever by inhaling contaminated aerosols or consuming contaminated animal products. Contaminated aerosols, which can contain C. burnetii, are generated during the processing of parturient fluids, placenta, or wool from infected animals [14]. While C. burnetii has been found in ticks, arthropod-borne transmission to humans is believed to play only a minor role in outbreaks [15]. Q fever is a global concern, often associated with cattle and sheep farming [16]. One of the largest documented outbreaks occurred in the Netherlands between 2007 and 2010, where over 4,000 individuals became ill and 14 died [17,18].
- Line 96: What is Argus and Hamas ticks? Would be Argas and Hyalomma (or Haemaphysalis)?
Agree: the names argas and hyalomma were removed and replaced with Haemaphysalis.
- In general, the introduction has several repeated parts that need to be better structured.
Agree: the introduction was proofread
- Line 114-121: Remove" We conducted the first examination of C. burnetii epidemiology in ticks collected from both small mammals and cattle. Our findings showed the presence of bacterial DNA in various organs of M. erythroleucus, a small mammal commonly found in the savannah. We also identified several species of ticks as carriers of the bacterium. The number of ticks collected from cattle was higher during the wet season compared to the dry season, and more tick species were identified during the dry season. Moreover, we found more species of ticks positive for C. burnetii DNA during the dry season than during the wet season.". This text is results, discusion or other and not a part of objectives of the study.
Agree: the paragraph was removed
- Line 125: Remove the term "Supplemental".
Agree: the word supplemental was removed
- Line 127-128: Ixodidae should not be italicized.
Agree: Ixodidae was italicized
- Line 130: Rename Scheme 1 as Figure 1. In the figure, Legends are not comprensible due to a poor resolution.
Agree: scheme 1 was renamed to figure 1. Line 125
- Line 159: "pi" must be in uppercase..
Agree: reversed to uppercase
- Line 170-179: All the abbreviation in this paragraph does not necessary. This is very well stablished by taxonomic rules.
Agree: all the abbreviations were removed
- Line 179: Remove the term "Supplemental"
Agree: supplemental was removed
- Line 180: Genus name in the first word of any paragraph does not be abbreviated.
Agree: the abbreviation removed in all the paragraphs
- Line 186: Laelapide should not be italicized.
Agree; the italicized was removed
- Tables may well be inserted in the text rather than as supplementary material. Properly insert and renumber the tables.
Agree: the tables are inserted in the text
- Line 187: Remove the term "Supplemental"
Agree: remove
- Line 192: Remove the term "Supplemental"
Agree: removed
- Line 199: Remove the term "Supplemental"
Agree: removed
- Line 200-204: All the abbreviation in this paragraph does not necessary. This is very well stablished by taxonomic rules.
Agree: all the abbreviations were removed
- Line 206: Remove the term "Supplemental"
Agree: removed
- Line 209: Genus name in the first word of any paragraph does not be abbreviated.
Agree: the abbreviation was removed
- Tables should not have horizontal lines in each row.
Agree: removed
- In Tables, scientific names should not be abbreviated, unless there are footnotes identifying the names. It is assumed that figures and tables are elements that must be understood independently of the main text.
Agree: all the abbreviations were removed
- Legends of figures and tables are poor described.
Agree: the legends are improved
- Line 213: This figure is not representative. A figure in columns with expressed confidence intervals made in the R statistical package would be much better represented.
Agree: the figure was removed
- Figures need to be renumbered.
Agree: the figure was removed
- Line 219: Genus name in the first word of any paragraph does not be abbreviated.
Agree: the abbreviation removed
- Line 236: Genus name in the first word of any paragraph does not be abbreviated.
Agree: the abbreviation removed
Comments on the Quality of English Language
Agree: English native speaker proof read the manuscript
Minimal refinement edits are required.
Agree: the manuscript was refined
Reviewer 2 Report
In the manuscript by Ohlopkova et al., the authors describe the prevalence of Coxiella burnetii in mammals and ticks from RG. They collected and analyzed a lot of data. Some presentation is not clear and there are some omissions. Detailed below are revisions that are needed.
Line 32 - why is 0.7% considered "high prevalence?"
Line 32-33 - disagreement between 0.7% and 0.07%. Which is it?
Lines 95-98 - what evidence of tick to human transmission is there. Reference 9 does not show this. Q fever is the disease in humans, not all mammals.
Line 123 - clarify if these are wild or farm animals
Table S1 - the percentages do not make sense. For example, 85 M. erythroleucus were tested, but only 7 were positive for Coxiella. That is not 63.6 percent. Equation 1 uses total number of animals in denominator.
Table S1 - if Coxiella was not present in M. natalensis, why is this listed in the table?
Line 201 - Either the number 544 or the calculated percentage is incorrect.
Line 210 - There are no data presented indicating 9620 ticks. Where did these come from and what species are they?
Figure 1 is not useful. These data would be presented better as a table.
All supplemental tables should be presented in the main text of the manuscript.
Author Response
Our findings significantly contribute to our understanding of the role of ticks in the epidemiology of C. burnetii in RG. This study represents the first analysis of C. burnetii epidemiology in ticks collected from both small mammals and cattle in RG. We employed a highly specific PCR method to detect bacterial DNA, ensuring accurate identification. Our results revealed that multiple tick species can harbor the bacterium. Interestingly, we observed a higher prevalence of C. burnetii in ticks collected from small mammals inhabiting the savannah. This finding suggests that ticks may play a significant role in the epidemiology of C. burnetii. Furthermore, we noted a greater abundance of ticks collected from cattle during the dry season compared to the wet season. Additionally, we observed more variations in tick species during the dry season compared to the wet season. Similarly, a higher number of tick species tested positive for C. burnetii DNA in the dry season compared to the wet season.
In the manuscript by Ohlopkova et al., the authors describe the prevalence of Coxiella burnetii in mammals and ticks from RG. They collected and analyzed a lot of data. Some presentation is not clear and there are some omissions. Detailed below are revisions that are needed.
Line 32 - why is 0.7% considered "high prevalence?"
Agree: the percentile was corrected
Line 32-33 - disagreement between 0.7% and 0.07%. Which is it?
Agree: the percentile was corrected
Lines 95-98 - what evidence of tick to human transmission is there. Reference 9 does not show this. Q fever is the disease in humans, not all mammals.
Agree: the references were updated and the text was added to show the evidence of tick to human transmission of C. burnetii. Lines 100 to 103.
Line 123 - clarify if these are wild or farm animals
Agree: revised
Table S1 - the percentages do not make sense. For example, 85 M. erythroleucus were tested, but only 7 were positive for Coxiella. That is not 63.6 percent. Equation 1 uses total number of animals in denominator.
Agree: the percentages were corrected
Table S1 - if Coxiella was not present in M. natalensis, why is this listed in the table?
Agree: the typo was corrected
Line 201 - Either the number 544 or the calculated percentage is incorrect.
Agree: all the number were checked and corrected
Line 210 - There are no data presented indicating 9620 ticks. Where did these come from and what species are they?
Agree: the data was added. Line 198-200
Figure 1 is not useful. These data would be presented better as a table.
Agree: the figure was removed
All supplemental tables should be presented in the main text of the manuscript.
Agree: all the tables are inserted in the text
Round 2
Reviewer 1 Report
The text still has small taxonomic nomenclature errors. Some terms like "Coxiella burnetii" (Line 147) should be italicized. Others like "Ixodid" (Line 206) should not be italicized. It is up to the authors only a detailed review of this aspect.
The text is understandable.
Author Response
The text still has small taxonomic nomenclature errors. Some terms like "Coxiella burnetii (Line 147) should be italicized. Others like (Line 206) should not be italicized. It is up to the authors only a detailed review of this aspect.
Agree: The text was revised again.
The text is understandable.
The authors wish to thank the reviewer for his positive view towards our manuscript.
Reviewer 2 Report
Critiques are adequately addressed.
Author Response
The authors wish to thank the reviewer for his positive view towards our manuscript.